# Long-period microseismicity reveals cryptic earthquake-triggered fluid activity can facilitate caldera eruptions

Zilin Song[1,2], Andrew F. Bell [2] ✉, Peter C. LaFemina [3,4], Sophie Butcher[2,5], Mario Ruiz [6], Stephen Hernández[6], Patricia M. Gregg [7] & Yen Joe Tan [1] ✉

Spatiotemporal correlations between moderate-to-large earthquakes and volcanic eruptions indicate that earthquake-induced stress changes can trigger eruptions. Triggering is rarely instantaneous, with time-dependent processes inferred to mediate magma ascent to the surface, such as fracture propagation or fluid pressurization. While various processes have been proposed, observational constraints on the specific triggering mechanisms remain limited. Here we integrate an earthquake catalogue, generated by machine-learning and template-matching techniques to enhance microseismicity detections, with geodetic data to show that seismic triggering of the 2018 eruption of Sierra Negra Volcano, Galápagos Islands, was facilitated by a cryptic phase of fluid activity. Following 13 years of near-continuous magmatic inflation, totalling > 6.5 m, stress changes induced by a moment magnitude 5.4 intra-caldera earthquake on 26 June 2018 did not immediately trigger magma release from the pressurized magma reservoir. Instead, long-period earthquake swarms revealed post-seismic fluid activity along reservoir-bounding faults in the northwestern caldera, locally promoting edifice failure and facilitating magma intrusion that initiated 8 hours later. These observations demonstrate that even when a pressurized reservoir experiences significant stress perturbation, cascading processes could potentially be essential to initiate volcanic eruptions.

Intermediate-to-large magnitude earthquakes are spatiotemporally correlated with volcanic eruptions[1–4]. This correlation can be explained by earthquake-induced changes in crustal static and/or dynamic stresses, coupled with magmatic processes that increase magma overpressure within the system[4,5]. Static stress changes are commonly invoked, as volcano-magmatic systems are often aligned with the local to regional stress field, parallel to the principal horizontal crustal stress ($\sigma_1$) (e.g., refs. 6–8). Under such stress conditions, a reduction in normal stress, or 'unclamping' of magmatic plumbing systems, allows for magma migration through crustal pathways, facilitating effusive to explosive eruption episodes[6,8,9,10]. Dynamic stress changes, induced by the passage of seismic waves through magmatic systems, are also hypothesized to induce a host of magmatic processes that increase magma overpressure, leading to system failure and eruption[5]. However, geophysical observations that shed light on the specific mechanisms underlying static and/or dynamic stress triggering of

[1]Department of Earth and Environmental Sciences, The Chinese University of Hong Kong, Hong Kong S.A.R., China. [2]School of Geosciences, University of Edinburgh, Edinburgh, UK. [3]Alfred Wegener Institute, Helmholtz Center for Polar & Marine Research, Bremerhaven, Germany. [4]Faculty of Geosciences, University of Bremen, Bremen, Germany. [5]British Geological Survey, Edinburgh, UK. [6]Instituto Geofísico, Escuela Politécnica Nacional, Quito, Ecuador. [7]Department of Earth Science & Environmental Change, University of Illinois at Urbana–Champaign, Champaign, IL, USA. ✉e-mail: a.bell@ed.ac.uk; yjtan@cuhk.edu.hk

eruptions are often unavailable, perpetuating ambiguity in the processes connecting earthquake triggers and volcano eruptions.

Sierra Negra Volcano in the Galápagos Islands, Ecuador, is one of the few examples of a basaltic caldera undergoing resurgence[11–13] and offers an exceptional site to investigate the interaction between a magmatic system, volcano-tectonic structures, and the volcanic edifice. The caldera hosts an intra-caldera fault system, the Trapdoor Fault system (TDF). This fault system is likely the reactivation of structures formed during caldera formation and is hinged in the northeast (Fig. 1). Reverse motion along the TDF has uplifted the C-shaped Sinuous Ridge by ~ 150 m above the caldera floor in the southwestern region of the caldera. Spatially correlated with the southwestern limb of the TDF is the Minas del Azufre hydrothermal area, a solfatara with a maximum temperature of ~ 200 ˚C and that crosses at least two segments of the fault (Fig. 1b). Hydrothermal activity here has resulted in alteration of the lava pile and hydrothermal mineralization[14,15]. Displacement on the TDF occurs by discrete earthquakes following meters-scale magmatic uplift centered in the caldera during pre-eruptive periods[13,16,17].

The three most recent eruptions in 1979, 2005, and 2018 were preceded several hours by earthquakes near and along the TDF. Specifically, the 22 October 2005 eruption began 2 hour and 48 minutes following a moment magnitude ($M_w$) 5.5 earthquake with surface rupture on the TDF[17,18], while the 13 November 1979 eruption initiated

1 hour and 15 minutes after a magnitude ($M$) 4.3 earthquake[19]. Although precise constraints on the earthquake source location are lacking for the 1979 event, the TDF remains the only volcano-tectonic structure that could have hosted the earthquake, as the caldera ring fault shows no evidence of recent displacement[17]. Therefore, compared to most reported cases of volcanic eruptions triggered by tectonic earthquakes, which typically act over timescales of days to weeks or longer (e.g., refs. 1, 6), the approximately one to three hour-long intervals between earthquakes on the TDF and eruption suggest near-instantaneous opening of magmatic pathways by static stress changes from TDF fault slip[12,20].

Unlike the 1979 and 2005 eruptions, the 2018 eruption of Sierra Negra was well-documented by densely distributed seismic and geodetic monitoring networks (Fig. 1), providing important high-resolution insights into the eruption initiation mechanisms. After 13 years of near-continuous magmatic inflation totalling > 6.5 m, a $M_w$ 5.4 earthquake occurred along the southern TDF at 09:15 (all times UTC) on 26 June 2018, resulting in at least 1.83 m of co-seismic vertical displacement of the Sinuous Ridge[12] (Figs. 1, 2a). Coulomb Failure Stress and normal stress changes promoted slip and/or opening on structures in the northwestern and northern segments of the TDF following the $M_w$ 5.4 earthquake[16,21]. An intense seismic swarm and contemporaneous surface deformation initiated 8 hours (~ 17:00)

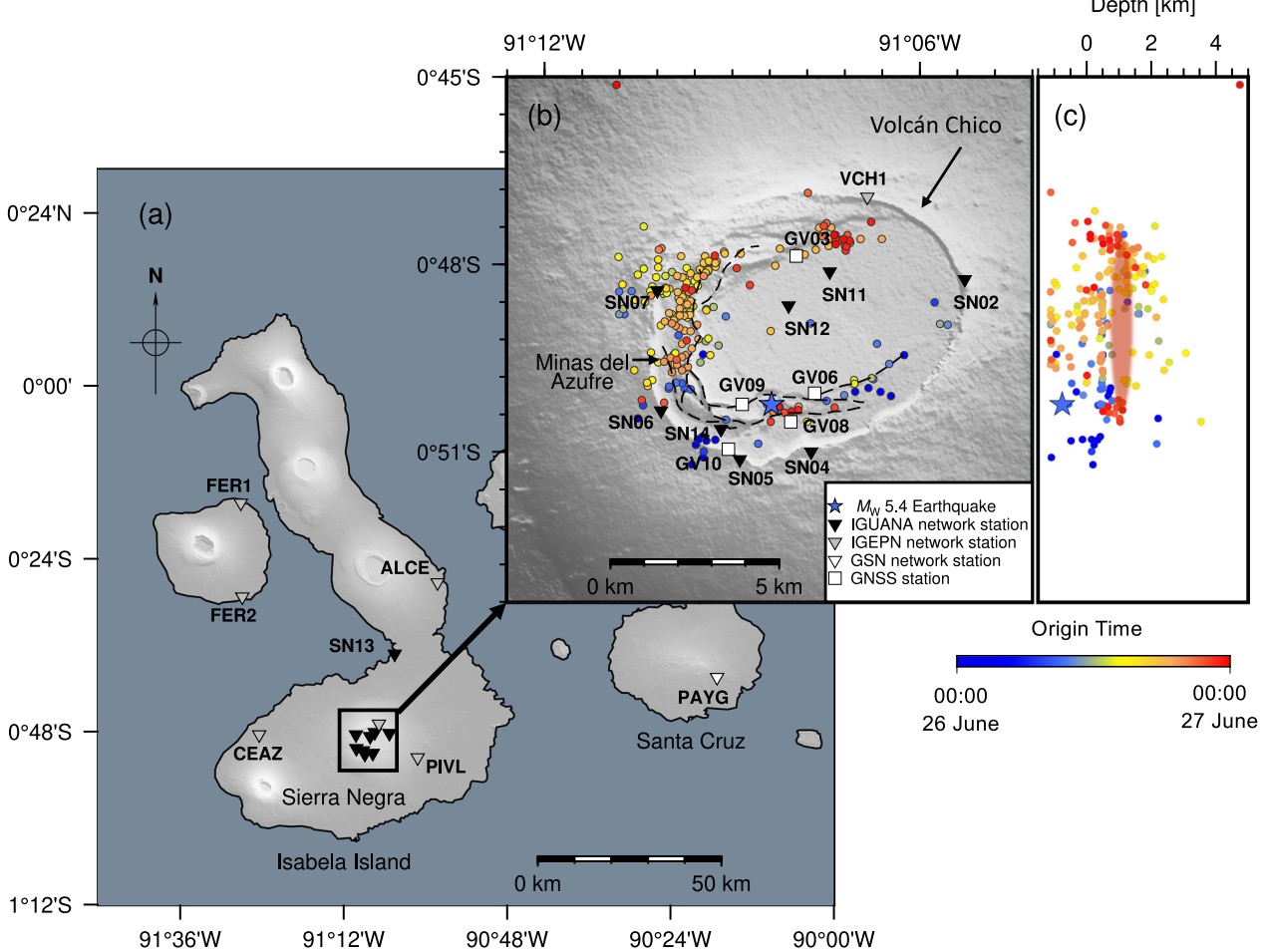

**Fig. 1 | Seismic and geodetic monitoring of the 2018 Sierra Negra eruption.**
**a** Map view of seismic stations (inverted triangles) within and around the Sierra Negra caldera, Galápagos Islands, used in this study. **b** The distribution of seismic events (circles) recorded on 26 June 2018[12], the Trapdoor Fault (dashed black lines), the seismic stations (inverted triangles), and continuous Global Navigation Satellite System (GNSS) stations (white squares) are overlain on topography[71]. The blue star

marks the epicenter of the moment magnitude ($M_w$) 5.4 earthquake that occurred at 09:15 UTC on 26 June 2018[12]. The circles and the blue star are color-coded by event origin time in UTC. The first eruptive fissures were located north-northwest of Volcán Chico[50]. **c** Vertical profile of earthquake hypocenters along a North-South cross-section. The sill-like magma reservoir is highlighted by the red-shaded zone[12].

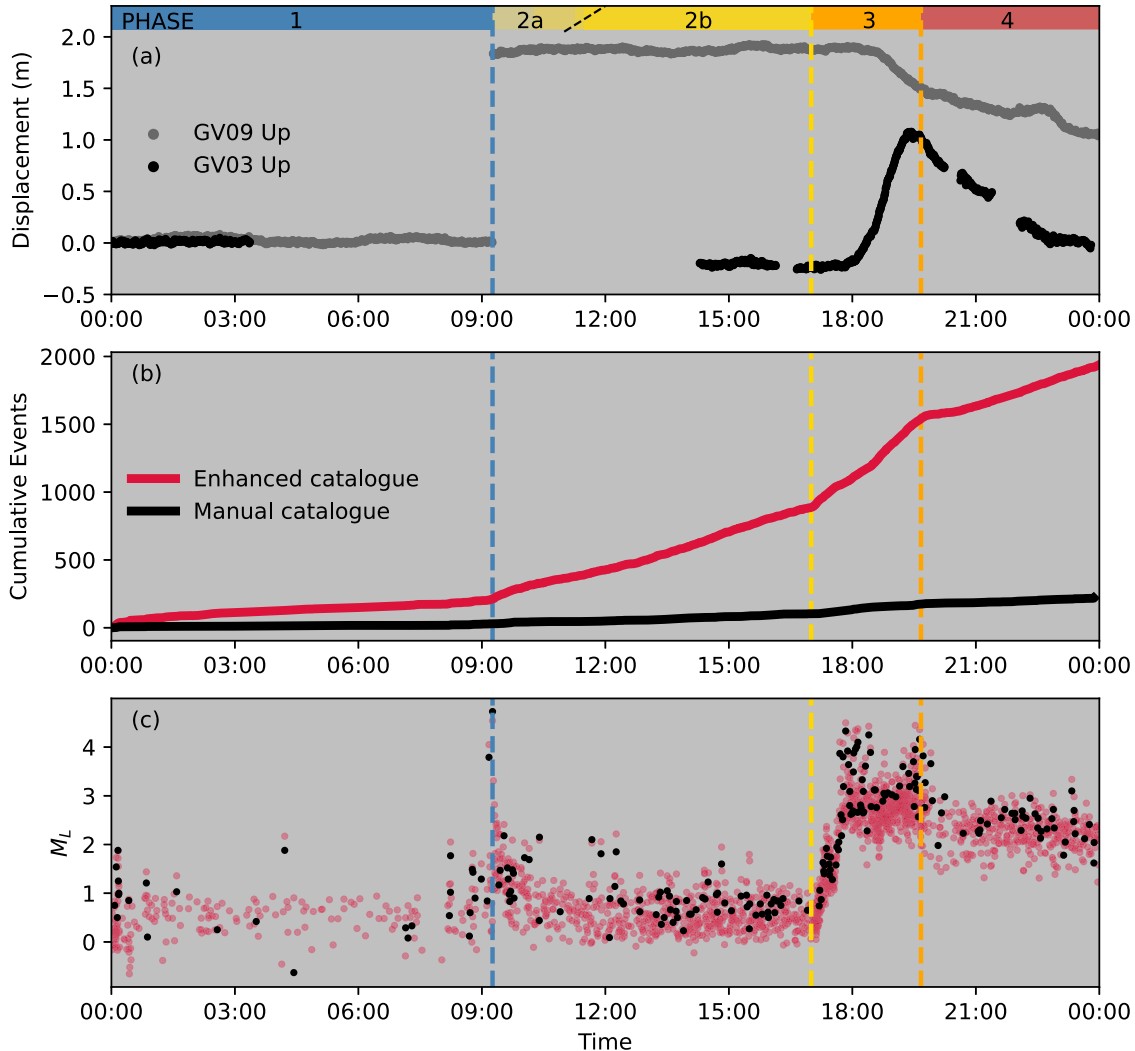

**Fig. 2 | Temporal evolution of seismic activity and surface displacement at Sierra Negra Volcano on 26 June 2018. a** Time series of surface deformation recorded by GNSS stations GV09 and GV03. Missing data for GV03 are due to a data recording issue in the GNSS receiver. **b** Cumulative seismic event counts from Bell et al.[12] (black line) and this study (red line). **c** Estimated local magnitudes ($M_L$) of earthquakes in Bell et al.[12] (black circles) and the enhanced catalogue of this study (red circles). Vertical dashed lines denote the timing of the $M_w$ 5.4 Trapdoor Fault earthquake (09:15; blue), magma intrusion (17:00; yellow), and eruption (19:40; orange).

after the $M_w$ 5.4 earthquake, indicating failure of the edifice surrounding the shallow magma reservoir and multi-directional magma propagation within the northwestern and northern caldera[12,22]. The emergence of intense seismic tremor at 19:35 marked the eruption onset[12,23]. In contrast to the onsets of the 1979 and 2005 eruptions, in which the eruptions occurred less than 3 hours after medium-to-large magnitude earthquakes ($M > 4$) on the TDF, the 2018 event exhibited an 8-hour delay between the TDF earthquake and initiation of magma migration, with eruption beginning 10 hours after the TDF earthquake. This delay raises critical questions regarding mechanisms governing the activation of magmatic activity after co-seismic stress perturbations.

The initial manually-picked earthquake hypocenter catalogue[12] documented seismicity preceding and during the 2018 Sierra Negra eruption (Fig. 1b). This catalogue predominantly comprised volcano-tectonic earthquakes (VTs) characterized by relatively high-frequency energy ($> 5$ Hz), indicating brittle failure within the volcanic edifice along the TDF[24]. However, despite the intense pre- and syn-eruptive seismicity, this catalogue lacks detail during the initiation phase, as manually-picked catalogues likely miss earthquakes with local

magnitude ($M_L$) < 1.0 (i.e., microseismicity[25,26]). Such microseismicity provides critical insights into magma-volcanic and/or volcano-tectonic processes that lead to eruption. This omission particularly affects detections of events with dominant low-frequency energy (1–5 Hz) and emergent phases, known as long-period earthquakes (LPs) that are closely related to magmatic and fluid activities[25,27–29].

In this study, we integrate machine-learning and template-matching techniques to generate a new high-resolution earthquake catalogue covering the period during the initiation of the Sierra Negra eruption on 26 June 2018. The resulting catalogue allows us to identify and track swarms of LP microseismicity in the northwestern corner of the caldera 2 hours after the $M_w$ 5.4 earthquake, which culminate in failure of the edifice surrounding the magma reservoir within the same region 6 hours later. The statistics and waveform properties of these LPs strongly suggest seismicity driven by cryptic fluid activity within the northwestern caldera ring fault and TDF region. These observations reveal cascading processes that led to eruption: co-seismic stress changes initiated localized fluid-driven stress perturbations, which eventually facilitated failure of the edifice and drainage of the sill-like magma reservoir.

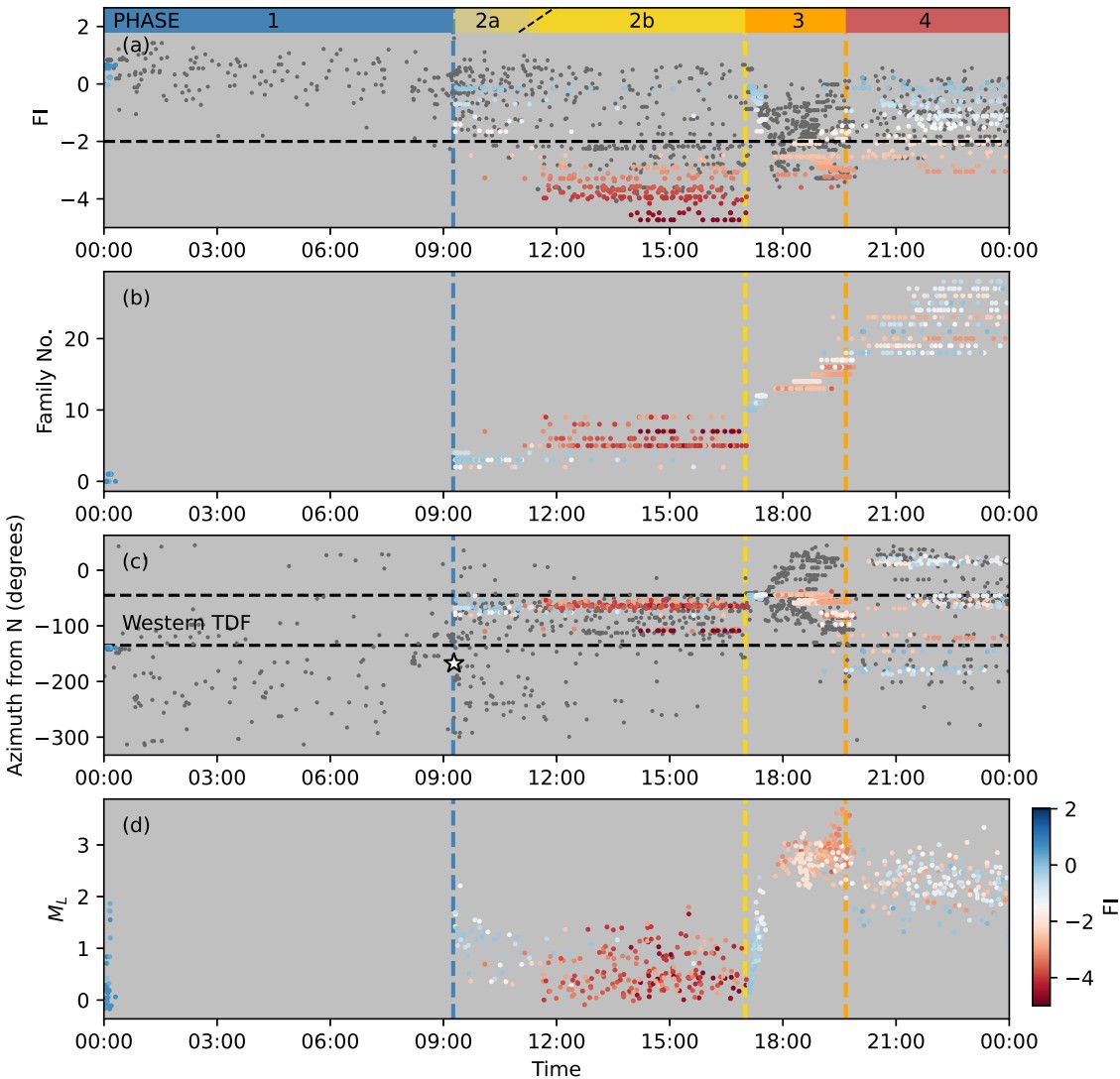

**Fig. 3 | Properties of the seismicity recorded at Sierra Negra on 26 June 2018. a** Temporal evolution of the frequency index (FI) for earthquakes in the enhanced earthquake catalogue (grey dots). The horizontal dashed line marks the FI threshold of −2. Earthquakes within multiplet families in a-d are color-coded by FI values. **b** Temporal distribution of identified multiplet families. **c** Azimuth distribution of earthquake epicenters relative to the caldera center (grey dots). Horizontal dashed lines delineate different segments of the Trapdoor Fault system. **d** Estimated $M_L$ of earthquakes within identified multiplet families. Vertical dashed lines in (**a**–**d**) indicate timings of the $M_w$ 5.4 earthquake (white star in **c**), magma intrusion, and eruption onset, as presented in Fig. 2.

## Results

### Seismicity and deformation associated with eruption initiation

We define four sequential phases of volcanic activity on 26 June 2018 at Sierra Negra caldera based on our analysis of seismic and geodetic data (Fig. 2). These phases are: continued pre-eruptive sill inflation (uplift) and seismicity culminating in the $M_w$ 5.4 TDF earthquake (Phase 1); post-earthquake aftershocks and pre-intrusion unrest (Phase 2); co-intrusive seismicity and deformation (Phase 3); and syn-eruptive subsidence and seismicity (Phase 4). Frequency index (FI) analysis[29–33] (see Methods) allows systematic classification of seismicity during these 4 phases into 1224 VTs and 711 LPs (Fig. 3a, Supplementary Fig. S1). Phase 1 exhibits sustained moderate rates of VTs in the vicinity of the TDF (Fig. 4a), concurrent with surface uplift at ~4 mm/day in the center of the caldera. At 09:15, a $M_w$ 5.4 earthquake induced a minimum 1.83 m uplift of the Sinuous Ridge along the southern segment of the TDF (Fig. 2a, c). This earthquake was immediately followed by a sequence of VT aftershocks located on the southern segment of the TDF, and repeating VTs north of Minas del Azufre, in the northwestern caldera (Figs. 3, 4b; Phase 2a). Two hours later, a new population of seismicity

appeared in the same region (Phase 2b), predominantly composed of LP microseismicity (Fig. 3a, d), which has not been previously reported in historical records from Sierra Negra or other Galápagos volcanoes (Fig. 1b, c). It is notable that LPs were not evenly distributed across the TDF, but spatially clustered in the northwestern caldera, north of Minas del Azufre (Figs. 1b, 3a, 4b). Throughout Phase 2, vertical displacement data from continuous Global Navigation Satellite System (cGNSS) stations across the caldera showed no significant changes above noise levels or signals that could be associated with magmatic or volcano-tectonic processes (Fig. 4e). Seismicity rates and magnitudes increased at 17:00 (Fig. 2b, c), marking failure of the edifice in the northwestern caldera. Magma evacuated from the sill and propagated bidirectionally into the northern and northwestern sectors of the edifice (Phase 3, Fig. 4c). Displacements in high-rate (30 s) cGNSS time series tracked magma migration in the northern caldera and progressive sill deflation starting at ~17:30[12] (Fig. 2a). The eruption commenced at approximately 19:40 (Phase 4), accompanied by relatively moderate rates of VTs across the caldera during the first few hours and rapid deflation of the caldera (Figs. 2, 4d).

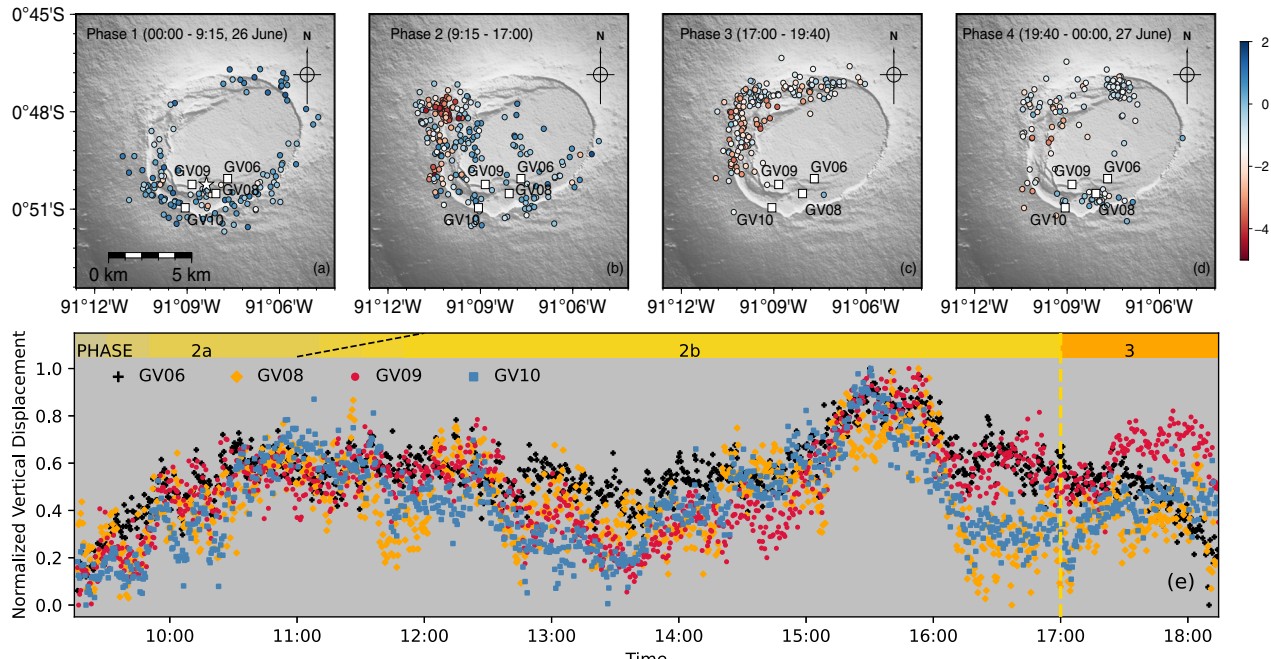

**Fig. 4 | Evolution of pre- and syn-eruptive seismicity on 26 June 2018.**
**a**–**d** Epicentral distributions of seismicity overlaid on topography[71] for the four phases of activity at Sierra Negra Caldera. Seismic events are color-coded by their FI. **e** Normalized vertical displacements of high-rate (30 s sampling) GNSS data recorded by continuous stations GV06, GV08, GV09, and GV10 during Phase 2 and the initial stage of Phase 3. Normalization of displacements is based on maximum displacement at each station, with values of - 14 cm for GV06 and - 10 cm for the other stations. The similarity in the amplitude of deformation across stations inside and outside of the caldera suggest noise in the high-rate solutions, and not magmatic or volcano-tectonic signals.

## Families of earthquake multiplets

Earthquake waveform clustering applied to the 26 June enhanced catalogue identifies 29 distinct earthquake families, each involving at least 10 multiplet earthquakes (see Methods), comprising a total of 714 earthquakes (Fig. 3b). Earthquake families are classified into LP and VT types by the average FI values of clustered events. Two short-lived (duration <20 minutes) VT families (average FI ≥ − 2) were detected in the early hours of 26 June along the southern segment of the TDF, where the fault ruptured 9 hours later, resulting in the $M_w$ 5.4 earthquake (Fig. 3). Three distinct VT families emerged concurrently and immediately after the $M_w$ 5.4 earthquake in the northwestern corner of the caldera (Phase 2a). Approximately 2 hours into Phase 2, this VT activity evolved into 5 LP families, located near the Minas del Azufre hydrothermal system and northwestern TDF (Figs. 1b, 3c) and characterized by average $M_L$ < 1 (Phase 2b; Fig. 3d). Multiplets within families exhibited repetitive occurrences with time intervals between successive events, defined as recurrence time ($T_r$), spanning 3-11 minutes on average. These events indicate localized stationary sources that rapidly reactivate over time[34,35]. The 5 LP families persisted for approximately 6 hours and ceased immediately with failure of the edifice surrounding the sill-like magma reservoir, marking the end of Phase 2 at 17:00[12]. Three short-lived VT families subsequently appeared on the northwestern corner of the TDF at the beginning of Phase 3 (Fig. 3). Their event magnitudes increased from 0.1 to 2.3, reflecting energy release potentially from dynamic magma intrusion[36–38] (Fig. 3d). This process was followed by 4 persistent LP and 1 VT families in the same region, which are characterized by an average $M_L$ of 2.8 (Fig. 3a, d). Throughout Phase 3, only one VT family originated on the southern TDF, persisting into Phase 4. No other earthquake families were identified on either northern or southern TDF before the eruption (Fig. 3c). The eruption commenced at ~19:40 with new VT and LP multiplet families appearing across all TDF segments, persisting at least until the end of the day, concurrent with rapid deflation of the central caldera (Figs. 2a, 3).

Repeating VTs in volcanic regions typically result from repeated brittle failure, slip, and healing of asperities[24], while the nature of volcanic LP multiplets requires non-destructive source mechanisms dominated by low-frequency energy, such as fluid activity[25,27,28] or slow rupture processes[39]. Frequency content perturbations from path or site effects (i.e., intrinsic seismic attenuation/scattering in the shallow edifice[40,41]) could play a role at Sierra Negra. However, we identify VT sources within 0.4 km of LPs (Supplementary Fig. S2), consistent with the typical location uncertainties for earthquakes involved in the catalogue (see Supplementary Methods). This observation suggests that the high-frequency energy deficit in LPs here is primarily related to source processes instead of path or site effects.

## Cryptic fluid activity as part of the eruption initiation process

Further insight into the nature of the LP source processes can be gained from the observation that 11 LP earthquake pairs occur as anticorrelated multiplets, also known as anti-repeaters (Fig. 5c and Supplementary Fig. S3a), based on waveform similarity (normalized cross-correlations (NCC) < − 0.7) and manual inspection (see Methods). These anticorrelated multiplets demonstrate systematic waveform polarity reversals, requiring rapid local stress reversals, which are commonly associated with pronounced local stress heterogeneity or fluid activities[42,43]. Interestingly, these anti-correlated multiplets are only identified during Phase 2, in LP families located within the northwest caldera. Anti-correlated LPs are not identified in other phases, and anti-correlated VTs are not detected at all throughout Phases 1–4. Therefore, the unique occurrence of anticorrelated LPs in the northwestern caldera prior to magma migration provides crucial evidence for rapid local stress variations[42], most likely associated with fluid-driven processes.

Robust positive correlations between seismic moment release ($M_O$) and $T_r$ ($R^2$ = 0.7, Fig. 5a) in VT families during Phases 3-4 indicate repeated brittle failure and stress reloading coupled with stick-slip instabilities[44] and fault creep[45], similar to observations at Mount Agung

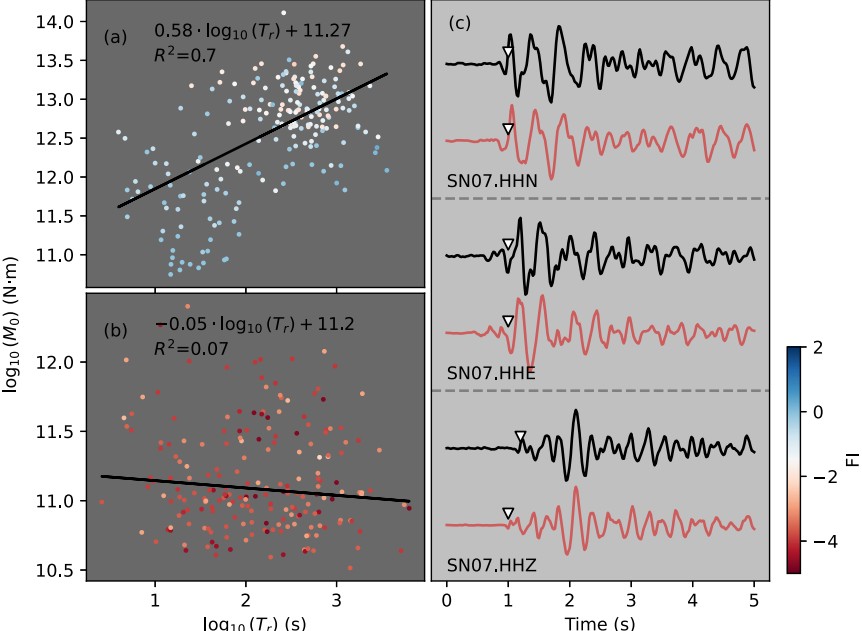

**Fig. 5 | Seismic moment ($M_O$) versus recurrence time ($T_r$) scaling and waveform characteristics of volcano-tectonic (VT) and long-period (LP) earthquakes.** **a** $M_O$-$T_r$ scaling for post-intrusion VT families. **b** Corresponding scaling for earthquakes in the 5 pre-intrusion LP families. Linear regressions (black lines) and event distributions (circles color-coded by FI values) are shown for both panels (**a**, **b**), with $R^2$ indicating the coefficient of determination for each regression. Statistical descriptions are detailed in the text. **c** Example waveforms of anti-correlated LP multiplets recorded by station SN07. Red and black traces represent the pre-filtered (2–20 Hz) original and polarity-reversed waveforms of paired multiplets, respectively. P- and S-wave arrivals, marked by white triangles, are derived from vertical and horizontal channels, respectively, and aligned by cross-correlation analysis between waveform pairs.

during the pre-eruptive phase[46]. However, LP families in Phase 2b lack statistically significant $M_O$-$T_r$ scaling ($R^2 = 0.07$; Fig. 5b), reflecting stress release processes distinct from classical quasi-static slip models, where fault strength recovery is proportional to the duration of stationary contact[35,44]. These findings eliminate slow rupture as a viable process for generating the LP signals[39] and provide further evidence that a fluid-related process is the source of LP multiplets, such as volatile-filled crack pressurization[47,48].

Periodicity of multiplet families designates sequences with values > 1 as exhibiting relatively lower variance in $T_r$ than expected for a Poissonian (temporally random) occurrence process (see Methods). Overall, the 5 Phase 2 LP families exhibit a maximum periodicity of 1, indicating that LP multiplets occurred randomly or slightly temporally clustered in time. In contrast, 12 of 13 Phases 3-4 VT families exhibit periodicity above 1, indicating the involvement of a temporal renewal process (Supplementary Fig. S4a). The lack of significant correlations ($r = 0.03$) between the average $T_r$ of multiplet families and periodicity excludes seismicity rate variations as confounding factors (Supplementary Fig. S4b). Collectively, these observations demonstrate that, unlike the potential periodic stick-slip processes identified by VT families, the 5 LP families in Phase 2b are most likely the result of a cryptic phase of fluid activity preceding the edifice failure surrounding the sill-like magma reservoir.

## Discussion

Previous studies have indicated global spatiotemporal correlations between tectonic earthquakes and volcanic eruptions, with the time interval from the earthquake to eruptions spanning from hours to years[2–4]. The correlation is hypothesized to be related to static and/or dynamic stress changes imparted on magmatic systems and resulting magmatic processes that increase magmatic overpressure. The last three eruptions of Sierra Negra also exemplified this correlation. The 1979[19] and 2005[17,18] eruptions of Sierra Negra Volcano were preceded less than 3 hours (i.e., 75 minutes and 168 minutes) by $M$ 4.3 and $M_w$ 5.5

earthquakes on the TDF, indicating near-instantaneous eruption triggering by static stress changes on the volcano-tectonic and magmatic systems (i.e., unclamping of faults and/or opening of magmatic conduits, magma migration from the sill-like magma reservoir at 2 km depth and eruption initiation). However, the volcano-tectonic complexity of the Sierra Negra system is exemplified by the history of local large earthquakes without eruption onset. For example, TDF earthquakes on 22 December 1999 ($M_w$ 5.0) and 16 April 2005 ($M$ 4.6) both occurred after prolonged magma accumulation in the sill, displaced the southern TDF and uplifted the Sinuous Ridge, but did not change the uplift rate in the caldera[17] (Supplementary Fig. S5). The 26 June 2018 $M_w$ 5.4 TDF earthquake generated comparable stress changes to the 1979 and 2005 events, promoting edifice failure in the northwestern caldera[12,21], yet the magmatic system response was delayed by ~8 hours. This delay suggests that even with considerable stress changes from TDF earthquakes on the volcano-tectonic system, additional cryptic processes are potentially required to ultimately trigger eruptions.

Integrating high-resolution observations of seismicity and surface deformation for the 26 June 2018 Sierra Negra eruption allows us to specify a multi-phase triggering process preceding the eruption (Phases 1-3; Fig. 6). At 09:15 on 26 June 2018, a $M_w$ 5.4 earthquake on the southern limb of the TDF produced > 1.83 m of co-seismic uplift at station GV09 on the Sinuous Ridge (Fig. 6a). This slip reduced the normal stress and increased Coulomb Failure Stress on faults (i.e., the ring fault and TDF systems) and fractures in the northwestern to northern caldera[12,21]. However, neither static nor dynamic stress changes proved sufficient to induce detectable changes in overpressure within the sill-like magma reservoir at a depth of 2 km (Figs. 2a, 4e). VT multiplet families initiated in the northwestern caldera instantaneously following the $M_w$ 5.4 TDF earthquake (Fig. 3). Swarms of LP microseismicity were then identified after 2 hours in the same region, indicating localized fluid activity (Fig. 6b). LP seismicity persisted for 6 hours before ceasing with a transition to VT seismicity

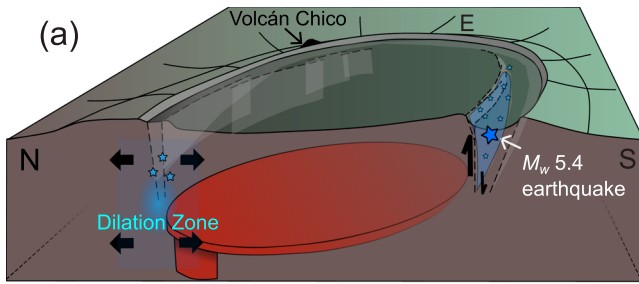

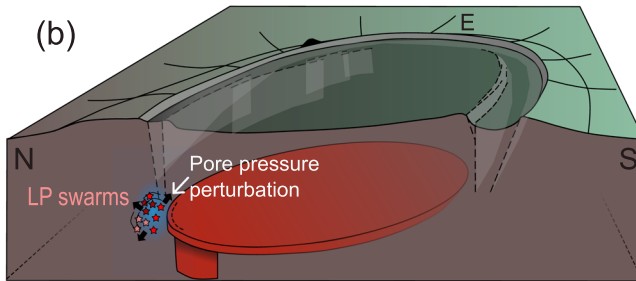

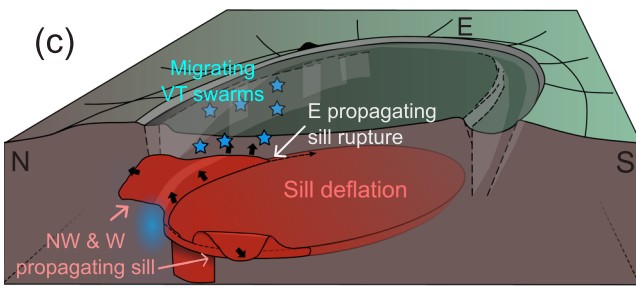

**Fig. 6 | Temporal evolution of seismic events and magma intrusion at Sierra Negra on 26 June 2018.** Cartoon summarizing observations following the 26 June 2018 $M_w$ 5.4 Trapdoor Fault earthquake during Phase 1 (**a**), Phase 2 (**b**), and subsequent magma intrusions during Phase 3 (**c**).

associated with failure of the edifice surrounding the sill and magma migration (Fig. 6c). This was followed by effusive eruptions, with fissures observed at 19:35 near Volcán Chico[49] (Fig. 1b).

In general, the consistent absence of both deformation and intense VTs across the caldera during the 8 hours after the $M_w$ 5.4 earthquake (Phase 2) excludes magmatic processes within the sill that would increase magmatic overpressure at 2 km depth (e.g., ref. 5; Figs. 2a, 4b, c, e). The lack of accelerating seismic occurrence rates during Phase 2 (Fig. 2b) further supports that progressive material weakening and failure mechanisms (e.g., ref. 51) are unlikely to be the dominant factors driving the edifice to failure. Interestingly, a key finding is that long-lasting (~6 hours) pre-intrusion LP multiplet families are spatially correlated with the location of magma migration initiation, offering further constraints on the underlying physical processes. The source-dominated low-frequency energy, unique presence of anti-correlated multiplets, non-periodic interevent times, and absence of $M_0$-$T_r$ scaling within these Phase 2 LP families, collectively indicate localized non-destructive processes most readily explained by fluid activity. High-temperature volatiles or fluids could change the friction coefficient along faults and fractures surrounding the reservoir by increasing pore fluid pressure[51] and promoting edifice failure and magma intrusion in the same area. We therefore propose that this cryptic phase of fluid activity plays an important role in weakening the volcanic system, facilitating edifice instability and eventually triggering the 2018 Sierra Negra eruption.

Intermediate-to-large magnitude earthquakes generate stress changes theoretically capable of initiating eruptions at volcanoes hosting primed magma reservoirs[20,51,52]. However, the context-dependent nature of volcano-tectonic systems, such as the local magmatic system[53], processes of volatile exsolution, bubble nucleation and migration[5], and pre-existing stress states[54], impact system responses to external triggers and modulates the response time between an earthquake and eruption. Unlike observations where static stress changes due to near-field large-magnitude earthquakes enhanced eruption potentials by decompressing magma chambers (e.g., Sumatra-Andaman arc[52]), increasing magma overpressures, and/or unclamping the magmatic plumbing system for dike migration from depth (e.g., Mount Fuji[6], Mauna Loa[52], and Cerro Negro[9]), pre-eruptive processes devoid of increased pressurization in the magma reservoir immediately prior to the 2018 Sierra Negra eruption offer critical insights into volcano-tectonic interactions. Given that previous ~M 5 earthquakes at Sierra Negra have triggered eruptions after delays of < 3 hours, while others have failed to trigger eruptions, the findings discussed above collectively emphasize that static stress transfer from large earthquakes alone may not uniquely be responsible for initiating failure in the edifice or structures near the sill-like magma reservoir and subsequent eruptions. Localized fluid processes, which promote magma intrusion through weakening fracture systems, could also play an important role in eruption initiation. This study provides critical implications for volcanic hazard assessments and motivates further studies of volcano eruption triggers.

## Methods
### cGNSS data analysis
Continuous GNSS data are processed following the methods in Bell et al.[12]. Specifically for this study, high-rate (1 epoch/30 s) kinematic positions are estimated using the precise point positioning (PPP) method implemented in the GIPSY-OASIS II version 6.3 software[55]. Kinematic analysis followed Larson and Miyazaki[56]. We apply ocean loading corrections using FES2004[57], and model wet and dry tropospheric zenith delays with VMF1 mapping functions[58]. We use the 30 s data to produce kinematic time series to investigate co-seismic displacements for pre- and syn-eruptive earthquakes and deformation related to magma migration. Co-seismic displacements are calculated by differencing the average of positions one hour before and after the earthquake. All data are available through the EarthScope archive (see Data Availability).

### Seismic data
To produce an enhanced earthquake catalogue, we use data from 15 three-component broadband seismometers located across the Sierra Negra caldera during 26 June 2018 (Fig. 1). Specifically, station VCH1 at Volcán Chico has been operating since 2012 as part of the 6-station Galápagos permanent seismic monitoring network run by the Instituto Geofísico, Escuela Politécnica Nacional (IGEPN)[59]. The 14-station IGUANA network, deployed on Sierra Negra in April 2018, recorded continuous seismic data from 9 operational stations during the 2018 eruption onset[12]. In addition, the seismic station PAYG on the neighbouring island of Santa Cruz, part of the Global Seismograph Network (GSN) since 1998, documented the seismicity associated with the onset of the 2005 eruption.

### Earthquake catalogue
We utilize Volpick, an EQTransformer-based phase picker for volcano seismicity developed by Zhong and Tan[33], to pick P- and S-arrival times for VTs and LPs across seismic stations on 26 June 2018. A total of 6261 P- and 5159 S-arrivals are subsequently associated into individual earthquakes through PyOcto[60] (see Supplementary Methods). With HypoInverse software[61] and HypoDD double-difference method[62], 694

earthquake hypocenters are successfully located with average location uncertainties determined to be 0.3 km and 0.4 km in horizontal and vertical dimensions, respectively (see Supplementary Methods). $M_L$ calculations for individual earthquakes and the determination of magnitude of completeness ($M_c$) for the earthquake catalogue are detailed in the Supplementary Methods.

To further enhance our catalogue, we apply a template matching algorithm through EQcorrscan[63], a Python package for cross-correlation-based earthquake detection. We first select 669 templates from the machine-learning based earthquake catalogue and cross correlate them with continuous seismic data recorded on 26 June 2018 (see Supplementary Methods). This process identifies 1241 new earthquakes, which are assumed to be co-located with their template events. Following Shelly et al.[64], we then estimate $M_L$ for template matching detections (M_Detection) based on relative magnitudes with respect to their associated templates (M_Template), using a scaling factor of 0.7 (see Supplementary Methods, Supplementary Fig. S6). With template-matching detections, the $M_c$ decreases by 0.25 for both Phases 1-2 and 3-4 earthquakes (Supplementary Fig. S7).

### Earthquake classification and clustering

Earthquakes are classified using pre-filtered vertical-channel seismograms (2–20 Hz), analyzed within 5 s time windows, starting 1 s before P arrivals at each station. We first label 694 well-located earthquakes as LP and VT by visually analyzing their velocity waveforms and spectral characteristics (see Supplementary Methods; Supplementary Fig. S1a, b). Considering potential subjectivity of manual labels[31], we reclassify earthquakes by further calculating FI of each earthquake following:

$$FI = \log_2(\bar{A}_{higher}/\bar{A}_{lower}) \qquad (1)$$

where $\bar{A}_{higher}$ and $\bar{A}_{lower}$ represent mean spectral amplitudes of seismic signals in high and low frequency bands at individual stations[30]. A scale of $\log_2$ provides an equal indication of the relative amount of high- and low-frequency energy within seismic signals by positive and negative FI values[32]. We use frequency bands of 9–15 and 2–5 Hz as they can differentiate 5 calibration LPs and VTs within the resulting earthquake catalogue reasonably well (Supplementary Fig. S1b). To mitigate station effects and systematic path attenuation during seismic wave propagation, we take the average FI of seismic signals with signal-to-noise ratio (SNR) $\geq 2$ as final estimates for individual events[31]. We find that manually-labeled events exhibit bimodal FI distributions, indicating that manually-labeled LPs show systematically lower FI compared to manually-labeled VTs (Supplementary Fig. S1c). This result is consistent with previous studies at Alaskan, Hawai'ian, and Japanese volcanoes[29,31,33]. We then applied the local minimum FI of −2 as a threshold to systematically classify events with FI below this value as LPs, which also signifies that low-frequency energy in LP signals is 4 times the high-frequency energy. Since only 47% of template-matching detection signals are well recorded (SNR $\geq 2$) which limits FI measurement, we thereby assume that all template-matching detections FI values are equivalent to their associated templates (Fig. 3a). This assumption is reliable as we do not observe systematic artefacts when comparing the assumed FI values to 1124 detections containing measured FI (Supplementary Fig. S8). Eventually, 1214 VTs and 711 LPs are identified within the enhanced earthquake catalogue.

Earthquakes exhibiting highly similar waveform characteristics are further clustered into families. Specifically, seismic signals with SNR $\geq 2$ in the 2–20 Hz frequency band are extracted from vertical channels for P and horizontal channels for S arrivals, with 5-s time windows starting 1 s before arrivals. We calculate the NCC for all event pairs at common stations, defining events as multiplets if their network-averaged NCC values exceed 0.7[34,65] (Supplementary Fig. S3). Multiplet events are further clustered into sequences if different multiplet event pairs share common events. Template-matching detections are assigned to the same sequence as their templates.

### Statistical methods

To investigate the underlying mechanisms for earthquake families, we then compare the characteristics of families between LP and VT multiplets. We first quantify the periodicity of each family as the ratio of the mean to the standard deviation of $T_r$, following Bell et al.[66]. Sequences with periodicity > 1 are defined as periodic, indicating a lower variance in $T_r$ than expected for a Poissonian occurrence process.

Multiplet families originating from stress release and reloading cycles during creep on rupture patches should theoretically exhibit positive correlations between $M_O$ and $T_r$[67]. To quantify this relation, we estimate $M_O$ of each multiplet event within families using the empirical conversion,

$$M_0 = 10^{1.5*Mw + 9.105} \qquad (2)$$

Where $M_w$ is defined as $M_w = M_L$ for $M_L > 3$[68] and $M_w = 2/3*M_L + 1$ for $M_L \leq 3$[69]. We perform linear regression on $M_O$-$T_r$ relations across the 5 pre-intrusion LP families (Phase 2) and post-intrusion VT families (Phases 3-4). To ensure statistical robustness in model fitting, we perform Jackknife resampling by systematically leaving out one sample at a time and repeating the analysis across all subsets (Supplementary Fig. S9). Our analysis suggests that VT families demonstrate significant positive $M_O$-$T_r$ correlations ($R^2 = 0.7$; Fig. 5a) consistent with slow-slip fault systems[70]. In contrast, LP families do not exhibit statistically significant scaling ($R^2 = 0.07$; Fig. 5b).

### Data availability

Seismic data recorded by the IGUANA network are available from EarthScope (10.7914/SN). Data recorded by the Ecuador seismic network (https://www.fdsn.org/networks/detail/EC/) can be accessed through the IGEPN website (https://www.igepn.edu.ec/datos-mseed). Geodetic data recorded by cGNSS stations for 26 June 2018 are available from EarthScope (https://www.unavco.org/data/doi/10.7283/3R4N-9775).

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

## Acknowledgements

This work was supported by the Hong Kong Research Grants Council General Research Fund (Grant No. 14300422 to Y.J.T.), the Croucher Tak Wah Mak Innovation Award (to Y.J.T.), and the Chinese University of Hong Kong PhD International Mobility for Partnerships and Collaborations Award (to Z.S.). Research on the 2018 eruption of Sierra Negra was supported by a U.S. National Science Foundation – U.K. Natural Environmental Research Council award (NSF-NERC 2122745 to P.C.L.) and (NE/W007274/1 to A.F.B.). S.B. acknowledges support from the British Geological Survey (BGS) International NC programme 'Geoscience to Tackle Global Environmental Challenges' (NERC: NE/X006255/1). The paper is published with permission of the Director of the British Geological Survey. Z.S. thanks Dr. Juan Zhu, Dr. Karen Lythgoe, Dr. Min Liu, and Yiyuan Zhong for helpful discussions.

## Author contributions

A.F.B. contributed to project administration, conceptualization, supervision, and seismic data collection. Y.J.T. contributed to methodology, supervision, and funding acquisition. Z.S. performed formal analysis and writing-original draft. P.C.L. contributed to geodetic data collection and analysis, and the article concept. All authors, including Z.S., A.F.B., P.C.L., S.B., M.R., S.H., P.M.G., and Y.J.T., contributed to the writing of the manuscript and have read and approved the final version.

## Competing interests

The authors declare no competing interests.
