## [Transparent Peer Review file · Nature Communications]

Long-period microseismicity reveals cryptic earthquake-triggered fluid activity can facilitate caldera eruptions

Corresponding Author: Professor Yen Joe Tan

Version 0:

Reviewer comments:

Reviewer #1

(Remarks to the Author)

This is a very well written and well conceived piece that is perfectly suited for Nature Communications. The observation of unusual earthquake activity in an unpredicted location during an interval between faulting and eruption is very important and yields insight into how eruptions might be nucleated from a pressurized magma chamber. This is an outstanding data set that is directly related to that problem.

I am not an experienced seismologist, but I found the explanations in the manuscript to be very clear, especially the changes in spectral signal. The authors do a good job of laying out all of the evidence that the unusual timing and frequency contents are likely driven by stresses in a porous water-saturated medium.

I have two suggestions that might improve the article:

1. A brief review of the hydrothermal system, especially the spatial distribution in relation to the seismic foci is in order. The best information comes from:

Goff, F., McMurtry, G.M., Counce, D., Simac, J.A., Roldán-Manzo, A.R. and Hilton, D.R., 2000. Contrasting hydrothermal activity at Sierra Negra and Alcedo volcanoes, Galapagos Archipelago, Ecuador. *Bulletin of Volcanology*, 62(1), pp.34-52.

Aiuppa, A., Allard, P., Bernard, B., Lo Forte, F.M., Moretti, R. and Hidalgo, S., 2022. Gas leakage from shallow ponding magma and trapdoor faulting at Sierra Negra volcano (Isabela Island, Galápagos). *Geochemistry, Geophysics, Geosystems*, 23(2), p.e2021GC010288.

2. I find the conclusions to be a bit vague, and to really get at the point the authors were making, I only figured it out by careful look at Figure 6. The text would be improved with a rhetorical summary. Something like:

1. Pressure from the subcaldera sill caused inelastic failure along the TDF.
2. Transient stress changes due to this slip caused pore pressure changes 2 km to the NW of the epicenter.
3. The pore pressures weakened the roof of the sill (or increased deviatoric stresses by reducing normal component), triggering the propagation of a dike from the sill.
4. How all this relates to the seismic measurements (what processes cause what signal)

I made a number of less-important suggestions as comments on the manuscript pdf.

Reviewer #2

(Remarks to the Author)

This is an interesting paper from an incredible set of observations. The technical work looks to be high quality and the paper is generally well organized and well written. While the interpretation that local fluid processes could help promote failure and magma intrusion is very reasonable, I don't think it is well supported. There is no attempt to understand the source mechanism for the earthquakes and the separation into distinct VT and LP categories oversimplifies the complexity here. Given that, I would argue that there is not enough support for the idea of a cryptic fluid being a part of the lead-up to the eruption. In short, I think this paper has great potential and with additional analysis, it could become a much more impactful

contribution.

To begin with, the FI data for the full dataset (Fig 3a) does not appear bimodal, which might make it easier to interpret VT and LP separately, but is instead continuous across the range. It would be interesting to see a histogram like that in Figure S1b for the full catalog to see. And even the data presented in S1b are weakly bimodal. With this dataset, it is very hard to determine which are VT and LP and so interpreting them as due to brittle failure or fluid involvement is very challenging.

First-motion mechanisms would go a long way toward helping explain the cause of the seismicity across the FI range. I realize this is a lot of additional work, but I see it as essential. It may be that many of the events with $FI < -2$ still have mixed first motions that could be explained by a double-couple mechanism. Furthermore, understanding the sense of slip on these faults would allow for comparison with the FEM modeling from Gregg et al.

While the paper is generally well written, there are sections of the paper that are not clear to someone who isn't familiar with all the literature cited about Sierra Negra volcano. It would be relatively easy to correct this.

For example, could you be more clear in the introduction about the TDF mechanism. Which way does the 'door' open? The trap door faults are not plotted on a map.

The term 'sill failure' is used multiple times. In descriptions of the sequence of events, I think this may not be appropriate since it is an interpretation. The idea of a large sill (e.g., Gregg et al., Bell et al.) is a model and is somewhat less definite than an earthquake hypocenter (also a model), so to refer to the sill failure as an observation, rather than an interpretation, seems incorrect. It is fine in the discussion, but what does 'sill failure' mean precisely?

The waveform shown in Figure 5 is interesting. It certainly looks to be dominated by relatively low frequencies. Based on the P wave, it seems like a nodal arrival. The black waveform has the picked P wave on the Z channel later than the first break on the N channel. Are these picks from EQTransformer or some other automated process? Also, it would make sense to remove the numbers on the vertical axis for part c since they are meaningless.

In the discussion of phase 2 beginning on line 269, the authors describe an absence of deformation and intense VTs. This does not seem well supported although it may be true. In Figure 4, that period is the only time with VTs in the caldera – though I agree there aren't many – and the GNSS data show deformation across all 4 stations plotted. The problem is that the displacement data are normalized so one can't tell what the scale of deformation is. I assume the reason for normalizing is to fit them all on one plot and show the consistency. In this case it would be helpful to have 4 scale bars showing the range of deformation represented by each of the stations.

In Figure 2, could you mention why there is missing data for GV03?

Regarding the caption for Figure 4, Phase 3 is mentioned as an 'eruption episode', but the eruption did not begin until 19:40 which marks the start of Phase 4

In Figure 5, the caption for part a should start with Seismic moment, not Moment magnitude

Line 212: I would argue that a R-squared of 0.07 is not a weak correlation, but is not a correlation at all

Line 245: I suggest a better term than 'near instantaneous' for something close to 3 hours

Reviewer #3

(Remarks to the Author)

Version 1:

Reviewer comments:

Reviewer #2

(Remarks to the Author)

As this is a re-review, I will keep my comments brief. I believe the paper is worthy of publication after the revisions. I commend the authors for the extensive work done to address the comments of the reviewers and feel satisfied with the responses to my comments.

Reviewer #1 (Remarks to the Author):

This is a very well written and well conceived piece that is perfectly suited for Nature Communications. The observation of unusual earthquake activity in an unpredicted location during an interval between faulting and eruption is very important and yields insight into how eruptions might be nucleated from a pressurized magma chamber. This is an outstanding data set that is directly related to that problem.

I am not an experienced seismologist, but I found the explanations in the manuscript to be very clear, especially the changes in spectral signal. The authors do a good job of laying out all of the evidence that the unusual timing and frequency contents are likely driven by stresses in a porous water-saturated medium.

Response: We have enhanced the text according to the comments raised (below), including clarifying terminologies and conclusions, to make the details of this study more readable, comprehensible, and robust.

1. A brief review of the hydrothermal system, especially the spatial distribution in relation to the seismic foci is in order. The best information comes from:

Goff, F., McMurtry, G.M., Counce, D., Simac, J.A., Roldán-Manzo, A.R. and Hilton, D.R., 2000. Contrasting hydrothermal activity at Sierra Negra and Alcedo volcanoes, Galapagos Archipelago, Ecuador. *Bulletin of Volcanology*, 62(1), pp.34-52.

Aiuppa, A., Allard, P., Bernard, B., Lo Forte, F.M., Moretti, R. and Hidalgo, S., 2022. Gas leakage from shallow ponding magma and trapdoor faulting at Sierra Negra volcano (Isabela Island, Galápagos). *Geochemistry, Geophysics, Geosystems*, 23(2), p.e2021GC010288.

Response: We have expanded the introduction of the Minas del Azufre hydrothermal system at Sierra Negra and its spatial correlation with the Trapdoor Fault system (TDF) on lines 73-77: *“Spatially correlated with the southwestern limb of the TDF is the Minas del Azufre hydrothermal area, a solfatara with a maximum temperature of ~200 °C and that crosses at least two segments of the fault (Fig. 1b). Hydrothermal activity here has resulted in alteration of the lava pile and hydrothermal mineralization (Aiuppa et al., 2021; Goff et al., 2000).”*

2. I find the conclusions to be a bit vague, and to really get at the point the authors were making, I only figured it out by careful look at Figure 6. The text would be improved with a rhetorical summary. Something like:

1. Pressure from the subcaldera sill caused inelastic failure along the TDF.
2. Transient stress changes due to this slip caused pore pressure changes 2 km to the NW of the epicenter.
3. The pore pressures weakened the roof of the sill (or increased deviatoric stresses by reducing normal component), triggering the propagation of a dike from the sill.
4. How all this relates to the seismic measurements (what processes cause what signal)

Response: Thank you for your constructive suggestion to clarify the key conclusions. We have revised the Discussion section (lines 300-307) to concisely outline the logical links between fluid activity and associated processes: *“The source-dominated low-frequency energy, unique presence of anti-correlated multiplets, non-periodic interevent times, and absence of M0-Tr scaling within these Phase 2 LP families, collectively indicate localized non-destructive processes most readily explained by fluid activity. High-temperature volatiles or fluids could change the friction coefficient along faults and fractures surrounding the reservoir by increasing pore fluid pressure (Caricchi et al., 2021) and promoting edifice failure and magma intrusion in the same area. We therefore propose that this cryptic phase of fluid activity plays an important role in weakening the volcanic system, facilitating edifice instability and eventually triggering the 2018 Sierra Negra eruption”.* The revised text emphasizes fluid activity as the primary driver of the observed seismic signals, while highlighting its cascading effects that contributed to triggering the 2018 eruption.

I made a number of less-important suggestions as comments on the manuscript pdf.

Line 47: I believe the pioneering paper is Nakamura (1977)

Response: We have included this citation in the manuscript (line 48).

Line 73: Perhaps clarify that the uplift is not centered on the TDF.

Response: We have clarified this point on lines 77-79: *“Displacement on the TDF occurs by discrete earthquakes following meters-scale magmatic uplift centered in the caldera during pre-eruptive periods (Bell et al., 2021a; Chadwick et al., 2006; LaFemina et al., 2025)”.*

Line 79: Kind of vague and I’m skeptical: what regional structures and how do they point to the TDF?

Response: As reflected in the revised text (lines 84-86): “Although precise constraints on the earthquake source location are lacking for the 1979 event, the TDF remains the only volcano-tectonic structure that could have hosted the earthquake, as the caldera ring fault shows no evidence of recent displacement (Chadwick et al., 2006)”, we have further specified the volcano-tectonic structures. The only fault structures at Sierra Negra capable of producing an earthquake of ~M5 are associated with the caldera. There is no indication that the outer ring fault has been active in the recent geologic history of the volcano.

Line 80: “the hours-long intervals between earthquakes on the TDF and eruption”

Response: We have revised the text to further specify the time intervals on lines 89-90: “...the approximately one to three hour-long intervals between earthquakes on the TDF and eruption...”.

Line 91: “failure along the margins of...” (a sill can’t really fail).

Line 177: “... failure of the sill” – see previous comments.

Line 273: Again, I don’t like this terminology, but perhaps it is ingrained. The sill never really fails. The wall rock does.

Line 275: change “failure” to “propagation”.

Line 262: In other words, the eruption was initiated by a sequence of mechanical disruption of the wall rocks instead of within the magma body itself.

Response: Since all these comments pertain to the terminology of "sill failure" used in our manuscript, we have consolidated them here for a unified response. We agree that “sill failure” can be potentially misleading. In reality, the sill itself doesn’t fail, instead, failure occurs in the edifice surrounding the shallow magma reservoir, which enables magma migration. Therefore, we have replaced the term “sill failure” with more precise phrases, such as “failure of the edifice surrounding the shallow magma reservoir” (line 101) and “failure of the edifice in the northwestern caldera” (line 150). Accordingly, we deleted "failure" from the original line 275 and changed it to “magma migration initiation” (line 299) to emphasize the spatial correlations between pre-intrusion LP multiplet families and magma initiation processes. We have reviewed the relevant terminology throughout the entire manuscript to ensure no ambiguity regarding "sill failure" remains.

Line 94-96: Perhaps restate the delay intervals for 79 and 05 for comparison

Response: Text on lines 103-106 has been revised: *“In contrast to the onsets of the 1979 and 2005 eruptions, in which the eruptions occurred less than three hours after medium-to-large magnitude earthquakes ($M > 4$) on the TDF, the 2018 event exhibited an 8-hour delay between the TDF earthquake and ...”*.

Line 245: I know it’s semantic but I wouldn’t call hours “near-instantaneous”

Response: The term ‘near-instantaneous’ here refers to the migration of magma from the sill in response to the triggering earthquake. We agree that “hours-interval” might not align perfectly with the strict meaning of “near-instantaneous” when evaluating the temporal link, at the first glance. However, as reflected in the text *“...global spatiotemporal correlations between tectonic earthquakes and volcanic eruptions, with the time interval from the earthquake to eruptions spanning from hours to years...”* (lines 259-260), from the perspective of the complete triggering process (i.e., tectonic earthquake → stress perturbation → edifice failure/magma migration → eruption), most such triggering processes typically operate on timescales of days to weeks or longer (e.g., Chesley et al., 2012; Bebbington & Mazocchi, 2011). Hence, by comparison, the approximately 3-hour windows between the $M > 4$ earthquake and volcanic eruptions at Sierra Negra in 1979 and 2005 would qualify as “near-instantaneous”.

As with the eruption in 2018, we envisage that a period of a few hours was subsequently required for that magma then to ascend the 2 km to the surface. To address the semantic concern, we have carefully clarified this definition in lines 87-91 (*“...compared to most reported cases of volcanic eruptions triggered by tectonic earthquakes, which typically act over timescales of days to weeks or longer (e.g., Bebbington & Mazocchi, 2011; Chesley et al., 2012), the approximately one to three hour-long intervals between earthquakes on the TDF and eruption suggest near-instantaneous opening of magmatic pathways by static stress changes from TDF fault slip (Bell et al., 2021b; Gregg et al., 2018)”*) and lines 265-269 (*“... eruptions of Sierra Negra Volcano were preceded less than 3 hours (i.e., 75 minutes and 168 minutes) by M 4.3 and M_w 5.5 earthquakes on the TDF, indicating near-instantaneous eruption triggering by static stress changes on the volcano-tectonic and magmatic systems (i.e., unclamping of faults and/or opening*

of magmatic conduits, magma migration from the sill-like magma reservoir at 2 km depth and eruption initiation”).

Bebbington, M. S., & Marzocchi, W. (2011). Stochastic models for earthquake triggering of volcanic eruptions. *Journal of Geophysical Research: Solid Earth*, **116**(B5), doi: 10.1029/2010JB008114.

Chesley, C., LaFemina, P. C., Puskas, C., & Kobayashi, D. (2012). The 1707 Mw8.7 Hiei earthquake triggered the largest historical eruption of Mt. Fuji. *Geophysical Research Letters*, **39**(24), doi: 10.1029/2012GL053868.

Line 279: I don't see how it is “stochastically activated”. It was activated by transient strain related to the quake

Response: We agree that the underlying process (i.e., fluid activity) was initially activated by the transient strain from the Mw 5.4 earthquake. However, the varying recurrence times of LP multiplets within identified families (with periodicity < 1) suggest that these activated processes subsequently drove LPs to recur stochastically. To address potential ambiguity, we have removed the expression and revised lines 301-303 as: “...and absence of M0-Tr scaling within these Phase 2 LP families, collectively indicate localized non-destructive processes most readily explained by fluid activity”.

Line 286: Gregg et al.

Response: Corrected (line 309).

Line 295: Clarify. There was increased pressurization over the course of years, but no sudden pressure change leading to eruption.

Response: We have revised the sentence on lines 318-320 to “...pre-eruptive processes devoid of increased pressurization in the magma reservoir immediately prior to the 2018 Sierra Negra eruption offer critical insights into volcano-tectonic interactions”. Thank you for clarification.

Line 297: A bit awkward. Maybe “The findings discussed above...”

Response: Text on line 321 has been revised as suggested: “...the findings discussed above collectively...” .

Reviewer #2 (Remarks to the Author):

This is an interesting paper from an incredible set of observations. The technical work looks to be high quality and the paper is generally well organized and well written. While the interpretation that local fluid processes could help promote failure and magma intrusion is very reasonable, I don't think it is well supported. There is no attempt to understand the source mechanism for the earthquakes and the separation into distinct VT and LP categories oversimplifies the complexity here. Given that, I would argue that there is not enough support for the idea of a cryptic fluid being a part of the lead-up to the eruption. In short, I think this paper has great potential and with additional analysis, it could become a much more impactful contribution.

Response: We appreciate the positive evaluation of the dataset, the technical work, and the manuscript. However, we respectfully disagree with two of the comments: 1) that the interpretation is not ‘well supported’; and 2) that source mechanisms would help the interpretation. To address these concerns, we have undertaken additional analyses, and revised the text, specifically with reference to other studies that support our approach and interpretation. We have added detailed explanations of methodologies in this letter and further interpretations in the manuscript, ensuring all the relevant points have been carefully treated.

To begin with, the FI data for the full dataset (Fig 3a) does not appear bimodal , which might make it easier to interpret VT and LP separately, but is instead continuous across the range. It would be interesting to see a histogram like that in Figure S1b for the full catalog to see. And even the data presented in S1b are weakly bimodal. With this dataset, it is very hard to determine which are VT and LP and so interpreting them as due to brittle failure or fluid involvement is very challenging.

Response: We agree that FIs in Fig. 3a exhibit continuous distribution of values. A pronounced bimodal FI distribution is not a reasonable expectation for dataset containing ‘LPs’ and ‘VTs’, nor is it necessary for defining event categories or critical to our interpretations. The frequency content of volcanic seismicity commonly displays a wide spectrum of properties, which depend on varying combinations of source processes and path effects. Similarly continuous FI distributions have been described

at Augustine volcano (Buurman & West, 2010) and Mount Agung (Wellik et al., 2021). Meanwhile, previous studies from volcanic settings in Japan (Zhong and Tan, 2024), Hawai'i (Matoza et al., 2014), and Alaska (Song et al., 2025, Augustine volcano: Buurman & West, 2010; Mount Agung: Wellik et al., 2021) have also used this "weakly-bimodal" FI distributions to categorize LPs and VTs. Key to this approach is the consistency between FI-based categorization and manual labelling. As noted by Matoza et al. (2014), volcanic earthquakes can be misclassified if one relies solely on subjective visual inspections of waveform characteristics. Therefore, although FI distributions are rarely distinctively bimodal, they provide an established, systematic, and objective means for earthquake classifications.

One of the crucial observations underpinning our interpretation of local fluid activation is that the seismicity in the northwestern corner of the caldera during Phase 2b has anomalously low FI values. In response to the question about FI values for the full catalogue, we have extended this analysis to include one month of seismicity before the eruption (published in Bell et al., 2021b). These data show rare occurrences of events with $FI < -2$, but with no spatial or temporal pattern or clustering (Figure R1). We would not be surprised if fluid-driven seismicity occurs in other locations or times at Sierra Negra, but this does not reduce the significance of seismic clusters in Phase 2b during the eruption initiation process.

In the revised manuscript, we have added sample waveforms of typical LPs and VTs in Fig. S1a (attached below) to help convey the FI analysis more intuitively, and revised the method on lines 386-389 (*“We find that manually-labeled events exhibit bimodal FI distributions, indicating that manually-labeled LPs show systematically lower FI compared to manually-labeled VTs (Fig. S1c). This result is consistent with previous studies at Alaskan, Hawai’ian, and Japanese volcanoes (Matoza et al., 2014; Song et al., 2025; Zhong & Tan, 2024)”*). We hope this can help address your concerns on the FI analysis.

Figure R1. Spatio-temporal distributions of FI for earthquakes that occurred one month before the day of the 2018 Sierra Negra eruption. (a) Temporal distribution of earthquakes (circles) that occurred one month before the eruption. Circles are colored by their FI values. The horizontal dashed line marks the FI threshold (-2) applied in this study to identify LPs. (b) Spatial distribution of earthquakes (circles) that occurred one month before the eruption. Circles are colored by their FI values. Gray circles are the seismicity identified by Bell et al., (2021b) at Sierra Negra caldera on 26 June 2018.

Supplementary Figure 1. (a) Sample waveforms of a calibration long-period earthquake (LP; red) and volcano-tectonic earthquake (VT; blue) recorded by the vertical channel of station SN05. Waveforms are filtered within the 2-20 Hz frequency band. To enhance microseismicity classification, we selected 5 largest-magnitude manually-labeled LPs and 5 largest-magnitude manually-labeled VTs (all magnitudes < 3) as calibration events. (b) Stacked spectra of 5 manually classified calibration LPs (red) and VTs (blue). Spectra for each event are calculated across stations with high-quality recordings (signal-to-noise ratio ≥ 2). The low and high frequency bands applied for frequency index (FI) calculation are highlighted by red and blue shaded regions, respectively. (c) Histograms of FI values for manually classified earthquakes, with red and blue colors representing LPs and VTs, respectively. The vertical dashed line indicates local minimum at $FI = -2$, defining the threshold for systematic earthquake classification.

Bell, A. F., La Femina, P. C., Ruiz, M., Amelung, F., Bagnardi, M., Bean, C. J., Bernard, B., Ebinger, C., Gleeson, M., Grannell, J., Hernandez, S., Higgins, M., Liorzou, C., Lundgren, P., Meier, N. J., Möllhoff, M., Oliva, S.-J., Ruiz, A. G., & Stock, M. J. (2021). Caldera resurgence during the 2018 eruption of Sierra Negra volcano, Galápagos Islands. *Nature Communications*, **12**(1), 1397, doi: 10.1038/s41467-021-21596-4.

Buurman, H., & West, M. (2010). Seismic precursors to volcanic explosions during the 2006 eruption of Augustine Volcano. *The 2006 Eruption of Augustine Volcano*. 1769.

Matoza, R. S., Shearer, P. M., & Okubo, P. G. (2014). High-precision relocation of long-period events beneath the summit region of Kīlauea Volcano, Hawai‘i, from 1986 to 2009. *Geophysical Research Letters*, **41**(10), 3413-3421, doi: 10.1002/2014gl059819.

Song, Z., & Tan, Y. J. (2025). Characteristics of deep long-period earthquakes at Alaska volcanoes from 2005 to 2017. *Journal of Geophysical Research: Solid Earth*, **130**, doi: 10.1029/2024JB030444.

Wellik, J. J., Prejean, S. G., & Syahbana, D. K. (2021). Repeating Earthquakes During Multiple Phases of Unrest and Eruption at Mount Agung, Bali, Indonesia, 2017. *Frontiers in Earth Science*, **9**, doi: 10.3389/feart.2021.653164.

Zhong, Y., & Tan, Y. J. (2024). Deep-Learning-Based Phase Picking for Volcano-Tectonic and Long-Period Earthquakes. *Geophysical Research Letters*, **51**(12), doi: 10.1029/2024GL108438.

First-motion mechanisms would go a long way toward helping explain the cause of the seismicity across the FI range. I realize this is a lot of additional work, but I see it as essential. It may be that many of the events with $FI < -2$ still have mixed first motions that could be explained by a double-couple mechanism. Furthermore, understanding the sense of slip on these faults would allow for comparison with the FEM modeling from Gregg et al.

Response: We understand why focal mechanism solutions (FMS) could be considered as an additional source of evidence to interpret the nature of the seismicity in Phase 2b. We did attempt to construct focal mechanisms for these earthquakes, but chose not to include them in the original submission. The low absolute magnitude of the events, low

signal-to-noise ratio (SNR), sparse station coverage, and emergent P-arrivals mean that the solutions are poorly constrained. However, we also argue that FMS are not particularly diagnostic of the underlying source process.

The average magnitude of LPs identified by FI in Phase 2 is 0.54, which is substantially lower than that of LPs used for focal mechanism analysis in previous studies. For example, Wech et al. (2020) stacked waveforms of LPs with magnitudes ranging from 1.2 to 1.5 to derive reliable focal mechanisms. Meanwhile, Bell et al. (2021) selected 80 events around the caldera with well-recorded P-wave polarities from at least 7 stations over a period of several months, but in our study, only 31 (8%) of the LPs in Phase 2 meet this threshold. Of these 31 LPs, only 22 are associated with 3 LP multiplet families in Phase 2b.

In response to these comments, we renewed our efforts to derive the FMS for these 22 LPs using P-wave polarities determined by QuakeMigrate (Winder et al., 2025). We then applied MTfit, a Bayesian approach to seismic moment tensor inversion (Pugh & White, 2018). We first analysed the source assuming fully double-couple (DC) mechanisms, and the results (Figure R2) suggest that, due to the challenges related to station coverage, the uncertainties of these solutions are poorly constrained in terms of the fault plane determination. Specifically, the maximum probability of DC solutions for the 22 LPs averages 0.1% with an averaged Kullback-Leibler divergence of 5.3. While the divergence provides a moderate concentration of the probability density function relative to a non-informative reference distribution, uncertainties of the derived FMS persist. We then analysed the data in the full moment tensor space. The Hudson plot (Figure R3) further reveals substantial uncertainties associated with DC mechanism assumptions, and specific source mechanisms remain poorly constrained.

Figure R2. Focal mechanism solutions of LP multiplets that are well-recorded ($SNR \geq 2$) by at least 7 stations within families No. 5, 6, and 9 during phase 2b. Stations with signals showing positive and negative polarities are indicated by white circles and black triangles, respectively.

Figure R3. Probability density functions of source types for 22 LP multiplets, which are plotted on the Hudson plot (Hudson et al. 1989). These LP multiplets were well-recorded ($\text{SNR} \geq 2$) by at least 7 stations, belonging to families No. 5, No. 6, and No. 9 during phase 2b. Several source types are shown: compensated linear vector dipole (CLVD) sources, implosive and explosive sources, opening and closing tensile crack sources (TC_+ and TC_-), and double-couple source (DC) sources. Light yellow indicates high probability, and dark purple indicates low (non-zero) probability.

We also tried to stack waveforms within each family to increase the SNR following Wech et al. (2020). However, we found that stacking waveforms did not consistently improve the SNR, particularly associated with the initial arrivals. In summary, the small amplitudes and emergent phase arrivals as well as poor focal sphere coverage render the FMS too poorly constrained to derive a reliable result.

Furthermore, we consider that even if they were available, reliable FMS would not be definitively diagnostic of whether the LP sources are fluid related. LP sources are known to exhibit both DC and non-double couple mechanisms (e.g., Oikawa et al., 2019;

Wech et al., 2020). Even for LPs with well constrained DC mechanisms, their potential association with the local stress field does not exclude the involvement of fluid activity in their source process (e.g., Oikawa et al. 2019).

Wech, A. G., Thelen, W. A., & Thomas, A. M. (2020). Deep long-period earthquakes generated by second boiling beneath Mauna Kea volcano. Science, 368(6492), 775-779.

Bell, A. F., La Femina, P. C., Ruiz, M., Amelung, F., Bagnardi, M., Bean, C. J., Bernard, B., Ebinger, C., Gleeson, M., Grannell, J., Hernandez, S., Higgins, M., Liorzou, C., Lundgren, P., Meier, N. J., Möllhoff, M., Oliva, S.-J., Ruiz, A. G., & Stock, M. J. (2021). Caldera resurgence during the 2018 eruption of Sierra Negra volcano, Galápagos Islands. Nature Communications, 12(1), 1397, doi: 10.1038/s41467-021-21596-4.

Pugh D. J., & White R. S. (2018). MTfit: A Bayesian Approach to Seismic Moment Tensor Inversion. Seismological Research Letters, 89(4), 1507–1513, doi: 10.1785/0220170273.

Hudson, J. A., R. G. Pearce, and R. M. Rogers (1989), Source type plot for inversion of the moment tensor, J. Geophys. Res., 94(B1), 765–774, doi:10.1029/JB094iB01p00765.

Oikawa, G., Aso, N., & Nakajima, J. (2019). Focal mechanisms of deep low-frequency earthquakes beneath Zao volcano, Northeast Japan, and relationship to the 2011 Tohoku earthquake. Geophysical Research Letters, 46, 7361–7370, doi: 10.1029/2019GL082577.

Winder, T., Bacon, C., Smith, J., Hudson, T., Greenfield, T., & White, R. (2020). QuakeMigrate: a Modular, Open-Source Python Package for Automatic Earthquake Detection and Location, doi: 10.17863/CAM.83083.

While the paper is generally well written, there are sections of the paper that are not clear to someone who isn't familiar with all the literature cited about Sierra Negra volcano. It would be relatively easy to correct this.

For example, could you be more clear in the introduction about the TDF mechanism. Which way does the 'door' open? The trap door faults are not plotted on a map.

Response: We have added text to the Introduction (lines 70-79) that includes further details on the description of Sierra Negra volcano: *“The caldera hosts an intra-caldera fault system, the Trapdoor Fault system (TDF). This fault system is likely the reactivation of structures formed during caldera formation and is hinged in the northeast (Fig. 1). Reverse motion along the TDF has uplifted the C-shaped Sinuous Ridge by ~150 m above the caldera floor in the southwestern region of the caldera. Spatially correlated with the southwestern limb of the TDF is the Minas del Azufre hydrothermal area, a solfatara with a maximum temperature of ~200 °C and that crosses at least two segments of the fault (Fig. 1b). Hydrothermal activity here has resulted in alteration of the lava pile and hydrothermal mineralization (Aiuppa et al., 2021; Goff et al., 2000). Displacement on the TDF occurs by discrete earthquakes following meters-scale magmatic uplift centered in the caldera during pre-eruptive periods (Bell et al., 2021a; Chadwick et al., 2006; LaFemina et al., 2025)”*.

We have also added the Trapdoor Fault in Figure 1b as requested (attached below).

Figure 1. (a) Map view of seismic stations (inverted triangles) within and around the Sierra Negra caldera, Galápagos Islands, used in this study. (b) The distribution of seismic events (circles) recorded on 26 June 2018 (Bell et al., 2021b), the Trapdoor Fault (dashed black lines), the seismic stations (inverted triangles), and continuous

Global Navigation Satellite System (GNSS) stations (white squares) are overlain on topography (Tozer et al., 2019). The blue star marks the epicenter of the moment magnitude (M_w) 5.4 earthquake that occurred at 9:15 UTC on 26 June 2018 (Bell et al., 2021b). The circles and the blue star are color-coded by event origin time in UTC. The first eruptive fissures were located north-northwest of Volcán Chico (Vasconez et al., 2018). (c) Vertical profile of earthquake hypocenters along a North-South cross-section. The sill-like magma reservoir is highlighted by the red-shaded zone (Bell et al., 2021b).

The term ‘sill failure’ is used multiple times. In descriptions of the sequence of events, I think this may not be appropriate since it is an interpretation. The idea of a large sill (e.g., Gregg et al., Bell et al.) is a model and is somewhat less definite than an earthquake hypocenter (also a model), so to refer to the sill failure as an observation, rather than an interpretation, seems incorrect. It is fine in the discussion, but what does ‘sill failure’ mean precisely?

Response: Based on comments from Reviewer 1, we have deleted the terminology ‘sill failure’ in the revised manuscript and have opted for more precise phrases, such as “failure of the edifice surrounding the shallow magma reservoir” (line 101) and “failure of the edifice in the northwestern caldera” (line 150), to avoid conceptual ambiguity and ensure the rigor of wording. We have reviewed the relevant terminology throughout the entire manuscript to ensure no ambiguity regarding "sill failure" remains.

The waveform shown in Figure 5 is interesting. It certainly looks to be dominated by relatively low frequencies. Based on the P wave, it seems like a nodal arrival. The black waveform has the picked P wave on the Z channel later than the first break on the N channel. Are these picks from EQTransformer or some other automated process? Also, it would make sense to remove the numbers on the vertical axis for part c since they are meaningless.

Response: As reflected in the text, “We utilize Volpick, an EQTransformer-based phase picker for volcano seismicity developed by Zhong and Tan (2024), to pick P- and S-arrival times...” (lines 354-355). The P- and S-wave picks annotated in Fig. 5c are derived directly from the phase picker. These waveforms are then aligned based on time shifts from cross-correlation analysis between waveform pairs.

We understand that the first break of the P-wave on the HHZ channel (black waveform) appears later than that on the HHE/N channels, which raised the question of a nodal

arrival. This discrepancy arises from the time alignment method applied in the figure: to highlight the characteristics of different phases, the vertical channel (HHZ) was aligned relative to the P-wave arrival while the horizontal channels were aligned relative to the S-wave arrival. Thus, the "0 s" on these channels do not represent the same absolute time, leading to the apparent delay of the P-wave on HHZ. This alignment strategy was intended to clarify phase characteristics but may have caused confusion. We therefore have added a note in the figure caption on lines 245-247 to explicitly explain this: "*P- and S-wave arrivals, marked by white triangles, are derived from vertical and horizontal channels, respectively, and aligned by cross-correlation analysis between waveform pairs*".

Also, as suggested, we have removed the meaningless numbers on the vertical axis for Fig. 5c. The updated figure is attached for reference.

Figure 5. (a) Seismic moment (M_0) versus recurrence time (T_r) scaling for post-intrusion volcano-tectonic earthquake (VT) families. (b) Corresponding scaling for earthquakes in the 5 pre-intrusion long-period (LP) families. Linear regressions (black lines) and event distributions (circles color-coded by FI values) are shown for both panel a and b. Statistical descriptions are detailed in the text. (c) Example waveforms of anti-correlated LP multiplets recorded by station SN07. Red and black traces represent the pre-filtered (2-20 Hz) original and polarity-reversed waveforms of paired multiplets, respectively. P- and S-wave arrivals, marked by white triangles, are derived from vertical and horizontal channels, respectively, and aligned by cross-correlation analysis between waveform pairs.

In the discussion of phase 2 beginning on line 269, the authors describe an absence of deformation and intense VTs. This does not seem well supported although it may be true. In Figure 4, that period is the only time with VTs in the caldera – though I agree there aren't many – and the GNSS data show deformation across all 4 stations plotted. The problem is that the displacement data are normalized so one can't tell what the scale of deformation is. I assume the reason for normalizing is to fit them all on one plot and show the consistency. In this case it would be helpful to have 4 scale bars showing the range of deformation represented by each of the stations.

Response: We have updated the figure caption to specify the normalization scale for deformation (lines 178-180): “*Normalization of displacements is based on maximum displacement at each station, with values of ~14 cm for GV06 and ~10 cm for the other stations*”. However, we hope to clarify that we do not share the view that the observation of a lack of deformation is insufficiently supported. The maximum amplitude of vertical displacement data from GNSS stations varies between 10~14 cm throughout Phase 2, and as stated in the text “... *showed no significant changes above noise levels or signals that could be associated with magmatic or volcano-tectonic processes...*” (lines 148-149). If this were a magmatic or volcano-tectonic signal, it would not have comparable amplitudes at stations within (GV06, 08, and 09) and certainly outside (GV10) of the caldera. To clarify our point that this is noise in the high-rate kinematic time series, we have added the following text to the caption for Figure 4: “*The similarity in the amplitude of deformation across stations inside and outside of the caldera suggest noise in the high-rate solutions, and not magmatic or volcano-tectonic signals*” (lines 180-182).

Figure 4. Evolution of pre- and syn-eruptive seismicity on 26 June 2018. (a-d)

Epicentral distributions of seismicity during the 4 phases of activity at Sierra Negra caldera. Seismic events are color-coded by their FI. (e) Normalized vertical displacements of high-rate (30 s sampling) GNSS data recorded by continuous stations GV06, GV08, GV09, and GV10 during Phase 2 and the initial stage of Phase 3. Normalization of displacements is based on maximum displacement at each station, with values of ~14 cm for GV06 and ~10 cm for the other stations. The similarity in the amplitude of deformation across stations inside and outside of the caldera suggest noise in the high-rate solutions, and not magmatic or volcano-tectonic signals.

In Figure 2, could you mention why there is missing data for GV03?

Response: We have added additional text in the Figure 2 caption stating: *“Missing data for GV03 are due to a data recording issue in the GNSS receiver”* (lines 158-159).

Regarding the caption for Figure 4, Phase 3 is mentioned as an ‘eruption episode’, but the eruption did not begin until 19:40 which marks the start of Phase 4

Response: We have revised the figure caption by deleting the words “eruption episodes” (line 178).

In Figure 5, the caption for part a should start with Seismic moment, not Moment magnitude

Response: We have corrected the typo in this figure caption (line 240).

Line 212: I would argue that a R-squared of 0.07 is not a weak correlation, but is not a correlation at all

Response: We have revised the text on lines 232-233 (*“LP families in Phase 2b lack statistically significant M_0 - T_r scaling ...”*) and line 419 (*“LP families do not exhibit statistically significant scaling ...”*) to ensure precise statistical clarification.

Line 245: I suggest a better term than ‘near instantaneous’ for something close to 3 hours

Response: We refer the reviewer to our response to reviewer 1 regarding the terminology used.

Reviewer #3 (Remarks to the Author):

Response: We would like to extend our sincere thanks for your participation in the collaborative review process and your dedication to supporting the fair and comprehensive assessment of our work.